# Technical and Methodological Aspects of Cell-Free Nucleic Acids Analyzes

**DOI:** 10.3390/ijms21228634

**Published:** 2020-11-16

**Authors:** Zuzana Pös, Ondrej Pös, Jakub Styk, Angelika Mocova, Lucia Strieskova, Jaroslav Budis, Ludevit Kadasi, Jan Radvanszky, Tomas Szemes

**Affiliations:** 1Institute of Clinical and Translational Research, Biomedical Research Center, Slovak Academy of Sciences, 845 05 Bratislava, Slovakia; zuzana.kubiritova@savba.sk (Z.P.); angelika.mocova@savba.sk (A.M.); ludevit.kadasi@uniba.sk (L.K.); 2Department of Molecular Biology, Faculty of Natural Sciences, Comenius University, 841 04 Bratislava, Slovakia; ondrej.pos@geneton.sk; 3Geneton Ltd., 841 04 Bratislava, Slovakia; lucia.strieskova@geneton.sk (L.S.); jaroslav.budis@geneton.sk (J.B.); 4Comenius University Science Park, Comenius University, 841 04 Bratislava, Slovakia; jakub.styk@gmail.com; 5Faculty of Medicine, Institute of Medical Biology, Genetics and Clinical Genetics, 811 08 Bratislava, Slovakia; 6Slovak Center of Scientific and Technical Information, 811 04 Bratislava, Slovakia

**Keywords:** body fluids, cell-free nucleic acids, exosomes, sample collection, extraction, storage, quantification, downstream analysis

## Abstract

Analyzes of cell-free nucleic acids (cfNAs) have shown huge potential in many biomedical applications, gradually entering several fields of research and everyday clinical care. Many biological properties of cfNAs can be informative to gain deeper insights into the function of the organism, such as their different types (DNA, RNAs) and subtypes (gDNA, mtDNA, bacterial DNA, miRNAs, etc.), forms (naked or vesicle bound NAs), fragmentation profiles, sequence composition, epigenetic modifications, and many others. On the other hand, the workflows of their analyzes comprise many important steps, from sample collection, storage and transportation, through extraction and laboratory analysis, up to bioinformatic analyzes and statistical evaluations, where each of these steps has the potential to affect the outcome and informational value of the performed analyzes. There are, however, no universal or standard protocols on how to exactly proceed when analyzing different cfNAs for different applications, at least according to our best knowledge. We decided therefore to prepare an overview of the available literature and products commercialized for cfNAs processing, in an attempt to summarize the benefits and limitations of the currently available approaches, devices, consumables, and protocols, together with various factors influencing the workflow, its processes, and outcomes.

## 1. Introduction

Cell-free nucleic acids (cfNAs), also called extracellular nucleic acids (ecNAs), represent fragments of DNA or RNA molecules that are actively or passively released from cells to different body fluids, including, but not limited to, blood, saliva, urine, or feces. Mechanisms of their release may be various, leaving typical molecular signatures for certain mechanisms and thus also for the physiological and pathophysiological process behind their release. Some cfNAs may be released passively following cell decomposition, for example by mechanisms such as apoptosis, necrosis, or physical damage, while others may be released actively to facilitate local or long-distance cell-cell communication, or even immune reactions, such in the case of NETosis or other similar processes. Their complex informational load, together with their general accessibility by relatively non-invasive sampling and associated possibilities of serial sampling, make them interesting analytes for several biomedical uses in a highly diverse range of medical fields. Possible applications may range from risk assessment and diagnostics, through monitoring of the course of the disease and treatment effectiveness, up to cfNAs involved in disease treatment [1,2,3,4].

The aims and utilities of cfNAs testing, therefore, may be different. It is possible, for example, to analyze the genome of the tested individual with an aim to characterize germline sequence variants from cfDNA fragments, however, nuclear DNA obtained from the cellular fraction of the biological material may offer more convenient ways to do this. If germline genomic variation is of interest from cfDNA, then it is obtained most commonly as a secondary product, such as retrieving population- specific or maternal genomic information from non-invasive prenatal testing (NIPT) data [5,6]. Germline genome characterization from body fluids of an individual may be, on the other hand, also performed with an aim to characterize the germline genome of other organisms, such as the fetal germline genome obtained from maternal plasma [7], from maternal saliva [8], or from the amniotic fluid [9], but also in a form of the identification of microorganisms from blood plasma, cervicovaginal secretion, or other body fluids [10]. When considering genome analyses, another high potential of cfNAs is represented by the characterization of somatic genome variations, for example, in tissues, organs, or even tumors, which may be yet undetected, or may be inaccessible by conventional methods. In addition, beyond genome analyzes, cfNAs may offer utility in the characterization of ongoing physiological or pathophysiological process, for example, through analyzes of either signaling molecules, most commonly small regulatory RNAs representing mediators of the processes, or molecules being products of the processes, such as DNA shed actively or passively into the body fluids.

Despite tremendous advances, however, cfNAs analyzes are still challenging, mostly because of issues such as contamination from post-sampling cell degradation, releasing further nucleic acids (NAs) to the sampled body fluid, but also because of the short half-life, low concentrations, or even high fragmentation of cfNAs [11,12]. Since such features are largely affected by preanalytical processing of samples, it is important to carefully choose and optimize protocols not only for the analyzes themselves, but also for sample collection, handling, transportation, storage, or cfNAs extraction and preparation [13,14]. Several studies address cfNAs processing methodology, however, the majority of them discuss cfDNAs, while only a few are dealing with different cfRNAs or microvesicles. Along with this, methodological aspects of the published studies are not always sufficiently informative, with several inconsistencies and missing information. In addition, the studies are characterized by differences in methodology from sampling to bioinformatic analysis, and to date, no standards have been agreed for study designs, making it difficult to cross-compare studies. Although review articles specialized to individual steps of cfNAs processing are available, even if they mainly focus on blood-based liquid biopsy, it seems that a comprehensive overview of this field is still missing.

Because of the abovementioned factors, we tried to collect available information about methodological aspects of analyzes of different types of cfNAs coming from several different body fluids. Using information retrieval from the available literature, and following web-based searching of relevant commercial products, we prepared the textual review, but also a graphical overview of the whole process (Figure 1), both of which are accompanied with five Appendix A summarizing the main information about certain aspects of cfNAs together with available products for their analyzes (Appendix A). It should be noted, however, that although our overview was prepared according to the best knowledge of the authors of the manuscript, and without any known commercial interest in the development and distribution of commercial biochemical reagents associated to cfNAs, the list of available commercial products should not be considered for absolutely comprehensive and other products may be available on the market.

## 2. Biological Material and the Utility of cfNAs Testing

When considering the origin and basic types of cfNAs, there are plenty of different types of cfNAs originating from various physiological and pathophysiological processes. Most basically, according to the type of the macromolecule, cfNAs are divided into cfDNAs or cfRNAs. In addition, endogenous and exogenous cfNAs can be differentiated, while endogenous cfNAs may be, for example, derived from nuclear (genomic) DNA (cf-ncDNA), mitochondrial DNA (cf-mtDNA), but also from a wide range of RNAs synthesized by the studied organism, whereas exogenous cfNAs are usually represented by, but are not limited to, food-borne, parasitical, viral, fungal or bacterial NAs [15,16,17]. Those originating from developing fetuses in maternal organisms may be, on the other hand, considered on the border between these two classes, since they originate from the maternal organism but belong to a different organism [4,15,16,17].

Although they are extremely dynamic in nature, and also various sampling, storage, and extraction methods may affect them, when considering the length of cfNAs, they were described in the range from ultrashort fragments, having tens of bps, up to fragments having several thousands of bps (Appendix A and references therein), while cf-mtDNA fragments tend to be shorter than cf-ncDNA. In general, cfDNA are mainly released following apoptosis, necrosis, or even NETosis, but their active release in a form of newly synthesized DNA associated to lipoproteonucleotide complexes and vesicles was also reported [18,19,20,21].

cfRNAs may be divided into many other types, including coding RNAs (mRNAs) and non-coding RNAs (ncRNAs), which encompass lncRNAs (long non-coding), small non-coding RNAs (sncRNAs), including miRNAs (micro), tRNAs (transfer), YRNAs, piRNAs (PIWI-interacting), circRNAs (circular), and other sncRNAs, such as ribosomal (rRNAs), small nuclear (snRNAs) or small nucleolar (snoRNAs). Among cfRNAs, the cf-mRNAs are fragmented and less abundant, therefore, many studies focus on the analysis of sncRNAs, which can generally be found encapsulated within vesicles or in a form of ribonucleoprotein complexes [1,22,23].

According to the form of their presence in the body fluids, both endogenous and exogenous cfNAs can be divided into free cfNA fragments (naked sequences having no specific vesicles), cfNA macromolecular complexes (e.g., nucleosomes, virtosomes, and neutrophil extracellular traps), or cfNA fragments encapsulated within extracellular membrane vesicles (EMVs; e.g., exosomes, microvesicles, and apoptotic bodies) [1,4,24].

Although there are body fluids containing cfNAs fragments from several organs and tissues, offering thus some kind of systemic information, a simple but general rule seems to be applicable, according to which body fluids may be significantly enriched with cfNAs derived from tissues, organs or organ systems being in the closest contact with the given fluid. However, the concentrations of cfNAs, their forms, length profiles, mechanisms of release, and their degradation are highly dependent both on the body fluids they are released to, as well as on the tissues they are released from, but also on the actual physiological and pathophysiological conditions of the organism (Appendix A). In addition, some body fluids can carry inhibitors, such as food digestion products, microbial contaminants, or other physiological components of the body fluid, which has the potential to affect downstream analytical processes [25]. Since each biological material is different in many aspects, the choice of an appropriate source should be carefully considered and should depend on the analyzed cfNAs, required applications, as well as on the aims of the analyzes. Blood, saliva, and urine appear to be the most commonly studied body fluids with respect to their cfNAs content, most likely because of their relatively easy sampling possibilities, large quantities available, but also because of their systemic content of cfNAs.

### 2.1. Most Commonly Assessed Body Fluids with Systemic Informational Potential

Currently, blood is the most commonly assayed source of cfNAs. While both serum and plasma can be used for cfNAs analyzes, although several comparative studies have shown serum cfDNA concentrations to be higher and fragment size to be longer compared to plasma. This is due to contamination by cellular NAs resulting from blood cell lysis after sample collection. Thus collection tubes and handling time are important factors for plasma, but especially for serum processing, suggesting that the plasma fraction may be more convenient for cfDNAs analysis [26,27]. Apart from cfDNA, cfRNAs are considered to be unstable molecules *per se*, but most of them were shown to form ribonucleoprotein complexes or were vesicle-bound. Several types of cfRNAs, including mRNAs, lncRNAs, and sncRNAs, such as miRNAs, piRNAs, circRNAs, tRNAs, rRNAs, YRNAs, snRNAs, and snoRNAs have been found in the blood, suggesting their potential as biomarkers for various diseases [1]. Exogenous cfNAs, extracted for example from the plasma, offer certain biomedical potential too, mainly in non-invasive identification of a wide range of infections [28]. Beyond the NAs belonging to the infectious agents themselves, infections may lead to specific molecular signatures even in endogenous cfNAs compositions, such as those resulting from cfDNA-induced vessel damage and cytokine storm [29].

Although salivary biomarkers concentration is generally lower than concentrations in blood, and therefore various enrichment procedures are crucial for successful downstream analyzes [30], accumulating evidence has proven that saliva has great potential for the detection and analysis of cfDNAs [30,31]. One of the greatest advantages is that saliva represents likely the most conveniently and comfortably accessible body fluid. Salivary cfNAs could originate from salivary gland secretion, epithelial cells turnover, oral microbiome, but also from blood circulation [32]. From RNAs, among others, mainly lncRNA, miRNA, piRNA, circRNA snoRNAs, tRNAs, snRNA were reported, being in macromolecular complexes with lipids, proteins, lipoproteins, and phospholipids, or encapsulated in vesicular structures that play a protective role against ribonucleases [33].

Urine has a great promise as a non-invasive sampling method for molecular diagnostics, since urinary cfNAs carry genetic information from cells shedding directly into the urine, but also contain fractions which may get to the urine through the glomerular filtration from the blood [17,34,35], including even cell-free fetal DNA (cffDNA) in pregnant women. However, since nuclease activity in urine was reported to be 4-fold higher than in plasma, fetal and maternal DNA in the urine are more degraded than those in plasma [36], accenting the necessity of proper and in-time processing of sampled material. In urine, cfDNA is composed of low-molecular weight DNA from the circulation or apoptotic cells of the urogenital tract and high-molecular weight DNA coming from necrotic cells in contact with urine (epithelial cells from bladder, prostate, kidney, etc.). The reason is that DNA in the blood is often present as a nucleosome, unable to be excreted into the urine, due to its size exceeding the pore dimension in the glomerular barrier [37]. Therefore, urinary cfDNA has been widely studied in urological diseases, such as kidney disease [34,35], tumors of tissues, and organs which may be in direct contact with urine [38]. It has been shown, for example, that cell-free tumor DNA (ctDNA) derived from renal cell carcinoma, bladder cancer or prostate cancer is detectable in more than 50% of plasma/serum samples while in more than 70% of urine samples of the same patients [39]. In addition to cfDNA, a number of biomarkers can be measured in urine, including circulating tumor cells (CTCs), cfRNAs (miRNA, lncRNA, and mRNA), proteins, and peptides, as well as exosomes [40].

### 2.2. Less Commonly Assessed Body Fluids Having rather Organ-Specific Informational Potential

Body fluids belonging to this class may, however, also bear systemic information, such as the detected cffDNA in maternal cerebrospinal fluid (CSF) [41], their availability for sampling, or their amounts available for sampling, make them rather suitable for more specific applications.

An advantage of stool testing is that it virtually assays the entire gastrointestinal tract, offering a promising source of cfNAs for screening and diagnosis of gastrointestinal and liver diseases, such as colorectal cancer, inflammatory, or infectious diseases [42,43] through changes of specific types of cfNAs, including cfDNA and cfRNA, for example cf-miRNAs, which may be released to the fecal matter [4,25]. To extend the range of biological processes reflected by cfNAs composition, regulatory RNAs, such as vesicle-bound miRNAs were found to be involved in specific targeting of bacterial genes, mediating thus direct interaction between the host organism and gut microbiota [44].

As CSF is in close contact with tissues of the central nervous system (CNS), including the spinal cord, brain, and cerebral ventricles [45], cfNAs presenting in this fluid may be informative about the processes of these otherwise strongly isolated tissues. In support of its potential, the higher concentration of cfNAs, such as ctRNAs and ctDNA derived from tumors of the CNS, was found in CSF when compared to blood plasma, most likely because of the strong reducing effect of the blood-brain barrier on CNS derived cfNAs in plasma [46,47,48,49]. Significantly more sensitive ctDNA-based tumor characterization was demonstrated from CSF, when compared to plasma, also with regard to resistance mechanisms on progression and monitoring of tumor response to treatment. When compared to blood plasma, CSF derived cfDNA was found to allow, for example, the detection of ALK receptor tyrosine kinase gene (ALK)-rearrangement in 81.8% of CSF samples, compared to 45.5% of plasma samples, of non-small-cell lung cancer patients having leptomeningeal metastases [50]. Beyond CNS tumors, specific CSF-derived miRNA subsets were reported to be highly accurate and discriminative even for diseases such as also in Parkinson’s disease and atypical parkinsonism [51], but also in patients following brain injury [52]. Apart from tumor DNA, cffDNA in the CSF of women during the peripartum period has been detected [41].

Since seminal plasma is a mixture of components secreted from the prostate, bulbourethral glands, testes, epididymis, and seminal vesicles [53], cfNAs content promise potential usability in diseases associated with these organs, such as male infertility and urologic malignancies, especially prostate cancer. cfDNA levels were, for example, found to be associated with sperm parameters such as quantity [54], viability, and alterations of motility and morphology [55], with significantly increased amounts of cfDNA in seminal plasma of prostate cancer patients [53]. In addition to cfDNA, several classes of cfRNAs, including ribosomal RNA, mRNA, miRNA, and piRNA molecules as single-stranded but also double-stranded forms have been described in semen [56]. Extracellular RNAs in human semen were shown to be stable because predominant cf-mRNAs were found in microvesicles, while most cf-miRNA were found to be bound with protein complexes [57].

Tear fluid, like saliva, represents an accessible but less complex body fluid, investigated mainly in the field of ocular surface disease [58], while excessive cfDNA was reported to be present in tear fluid of patients with dry eyes compared to healthy subjects [59]. Interestingly, tears of healthy individuals contain a large number of miRNAs, which may suggest an important role for these regulatory RNAs in maintaining or regulating the normal function of the eyes [60].

While cfDNA was identified in sputum samples of lung cancer patients too, with its amount to be in relation with the amount of inflammation, no differences in cfDNA concentrations between lung cancer patients and controls were found, even if the presence of tumor-related DNA was proven by methylation analysis [61]. On the other hand, the clinical value of sputum cfDNA for detecting driver genes alterations in non-small cell lung cancer (NSCLC) was confirmed, especially for smoking patients, while the combination of various body fluids, including sputum, plasma, and urine, improved the clinical utility of liquid biopsy in NSCLC [62].

Abundant and high-quality cfNAs derived from breast cells, such as lactocytes and myoepithelial cells, were reported to be present in breast milk in a relatively stable form, suggesting that breast milk might be a valuable non-invasive sample source for finding early alterations of NAs associated with the initiation and progression of breast cancer [63]. Beyond oncology, exosomes with RNA content, including mRNA, were identified in colostrum and mature human breast milk, with proven capacity to influence the immune system of the infant by vertical shuttling of regulatory cfRNAs via exosomes between individuals during breastfeeding [64,65]. In addition, a wide range of miRNAs were identified in breast milk, including novel ones, predicted to target genes which were found to be enriched for transcriptional regulation of metabolic and immune responses, with specific miRNA profiles being alterable by maternal diet [66]. These findings seem to outline the certain utility of cfNAs profiling of breast milk during breastfeeding, with the subsequent potential of modulation of fetal metabolism and immune system maturation through maternal dietary changes.

Although research of cell-free fetal nucleic acids (cffNAs) has primarily focused on maternal plasma, in line with the above delineated rule of body fluids becoming enriched with cfNAs derived from tissues and organs being in closest contact with the fluid, cffNAs might be found in much greater concentrations in the amniotic fluid, representing a pure fetal sample uncontaminated by maternal- and trophoblast-derived NAs. Technical progress has also made it possible to extract cffNAs, both cffDNA, and cffRNAs, of sufficient quality and quantity to perform functional genomic analyzes from amniotic fluid [9,67]. Moreover, a large number of detectable miRNA species were reported to be present in amniotic fluid, having unique composition when compared to other body fluids, suggesting that the filtering process by the placenta reduces the exchange between amniotic and other body fluids [60].

cfDNA in the supernatant of pleural effusion (PE) is a valuable source that can be used to detect the driver as well as resistance mutations and can guide tyrosine kinase inhibitor treatment decisions [68]. High accordance in mutational spectra among matched cfDNA from plasma, malignant PE, and ascites collected from advanced gastric cancer patients have been discovered [69]. There is also evidence supporting the clinical utility of exosomal DNA from malignant PE of advanced lung adenocarcinoma patients. Targeted MPS analysis of PE-derived exosomal DNA and cfDNA exhibited highly concordant mutational profiles [70]. In addition to cfDNA also miRNA molecules have been extracted from the pleural fluid [60].

Likely because of the excess fluid accumulating inside the peritoneal cavity is a result of fluid leakage from blood vessels, as a consequence of various diseases, fragment sizes of ascites cfDNA were found to correlate with those seen in plasma. Analyzes of cfDNA from the ascites of individuals with metastatic cancer demonstrated the presence of tumorigenic CNVs in cancer-associated genes [71], while mutation profiles of ascites derived cfDNA were also found to be highly concordant with cfDNA from plasma and even malignant pleural effusion in gastric cancer patients [69].

Liquid biopsy-based on bronchoalveolar lavage fluid (BALF) sampling has rarely been used, however, the molecular testing method for the detection of EGFR mutations in NSCLC patients using cfDNA from this source demonstrated high diagnostic value with 91.7% concordance with tumor biopsy results in the detection of EGFR mutations [72]. Intact RNA and miRNA were detected in cell-free supernatants of bronchial lavage fluid [60]. Moreover, similarly to other fluids, RNA concentrations were shown to be higher in lavage than in the serum samples of lung cancer patients [73].

Modern intravitreal chemotherapeutic techniques for retinoblastoma involve aspiration of aqueous humor, providing a novel sample source for analysis. Berry et al. demonstrated measurable concentrations of DNA, RNA, and miRNA in such samples [74]. Analysis of cfDNA from the aqueous humor fluid of retinoblastoma eyes was capable of detecting somatic variants and the results were comparable to current tests on enucleated tumor tissue [75]. Dysregulation of specific miRNAs in the vitreous humor might also be related to the alterations that characterize patients affected by the macular hole and epiretinal membrane [76].

There are also other body fluids in which cfNAs were assessed and evaluated, but studies for which are even scarcer, when compared to the abovementioned fluids. The presence of cfDNA in sweat, for example, was detected in 80% of tested healthy individuals, with an average concentration of 11.5 ng/mL [77]. Other body fluids and their cfNAs content are generally reported with respect to certain associated diseases. It was demonstrated, for example, that pancreatic juice is a useful source for genetic profiling of somatic alterations of pancreatic tumors and intraductal papillary mucinous neoplasm with a potential to assess also their malignant progression risk [78]. Since cervicovaginal secretion has close contact with certain types of tissues, it can be a useful specimen of genetic material for testing pathogens associated with sexually transmitted diseases or cervical cancer [10]. Somatic mutations, including single nucleotide variants (SNVs) and copy number variants (CNVs), were found to be detectable also within cfDNA obtained from the bile of biliary tract carcinoma patients. Interestingly, bile cfDNA was predominantly composed of long DNA fragments in contrast to plasma [79]. cfNAs content of synovial fluid was also assessed, while elevated cfDNA levels in the plasma and synovial fluid of rheumatoid arthritis patients were reported, with a median cfDNA concentration being ~77-fold higher in the synovial fluid than that in the plasma of rheumatoid arthritis patients [80]. Potential role in diagnostics and prognosis of therapy-related peritoneal membrane degeneration was assigned to cfDNAs coming from the peritoneal effluent of peritoneal dialysis patients [81]. The quantification of peritoneal cfDNA could be an innovative method to determine acute damage and an inverse index of the repair process [82]. In peritoneal dialysis patients, also miRNA molecules were extracted from cell-free effluent samples, while specific miRNAs were present in a fraction of microvesicles and exosomes [83]. In addition to these body fluids, several types of cyst fluids were shown to be valuable sources of cfNAs and having potential clinical diagnostic utility, including cfDNA and miRNAs from pancreatic cyst fluid of pancreatic cancer patients [84,85,86], or ovarian cancers with a potential to provide information about whether the cysts releasing them are benign or cancerous, but also about the presence of tumor-specific mutations of borderline tumors, type I cancers, and type II cancers [87].

## 3. Sample Collection

Although there are highly specialized and mature analytical technologies able to characterize cfNAs, to ensure unbiased analyzes, preanalytical processes need to preserve the original state of cfNAs as much as possible. cfNAs could both be diluted by the post sampling release of cellular NAs, but also by their degradation. Compared to cfDNA, cfRNAs are, for example, very unstable due to RNase activity, unless they are protected by vesicles or macromolecular complexes, in which they were reported to be stable in several body fluids [88]. The first one of these preanalytical processes is sample collection, during which it is important therefore to secure at least the following points: (i) to prevent degradation of free cfNAs fragments; (ii) to stabilize cells to prevent the undesired release of NAs from them; and (iii) to stabilize EMVs and macromolecular complexes to protect their cfNAs cargos and contents. Many devices or solutions were developed for the stabilization and preservation of cfNAs in body fluids, however, the most evaluated are devices designed for the preservation of cfNAs in blood. Some of the available preservation solutions, dedicated for cfDNA, prevent cell lysis and reduce the releasing of non-vesicular miRNAs, however, they are not sufficient for the preservation of constant concentration of EMVs after blood collection [14], since the time-dependent increase of cfRNA concentration was observed in the majority of these specialized collection tubes [89]. Among the wide range of body fluids, blood is the most commonly used and studied, for which a wide range of stabilization solutions are available, while recently they were developed also for saliva or urine (Appendix A). With the increasing use of other body fluids for liquid biopsy, collection material and stabilization solutions dedicated for these should be developed. Until that time, collection of other body fluids need to rely on collection materials developed for the same body fluid but for the stabilization of total NAs, dedicated to other body fluids, or even on home-made solutions, all of which may, however, deteriorate the results unless their thorough evaluation and validation are performed.

### 3.1. Blood

The volume of blood needed for the cfNAs analysis usually ranges between 6–10 mL, however, it depends upon the testing methodology. Commonly used collection tubes with anticoagulants such as di- or tripotassium ethylenediaminetetraacetic acid (K_2_EDTA, K_3_EDTA), lithium heparin, or sodium citrate can stabilize cfDNA only when plasma is processed within six hours, with EDTA performing slightly better than the other two at six hours and longer times [90]. For EDTA there were different time-frames reported for the preservation of cfDNA at room temperature, from a few hours up to 24 h [26,91,92,93] or even up to 48 h [94] over time the lysis of blood cells was detected. Over that time, the storage of blood in EDTA at 4 °C seems to delay the lysis [91,92,95] up to three days [26]. Stability of cf-miRNA up to 12, or even 18 h, was also reported in EDTA tubes at room temperature with higher stability at 4 °C [96]. In general, however, it is recommended to process the blood samples, when collected either into EDTA, lithium heparin, or sodium citrate, immediately after collection, or at least as soon as possible [12,94,97,98].

To prevent cell lysis when storing for a longer time, or during shipment to the laboratory, reagents such as formaldehyde or glutaraldehyde may be added, which, however, may cause damage of NAs due to cross-linking with proteins [99,100,101]. The best choice is to use specially designed collection tubes containing dedicated stabilization reagents, although the composition of which is generally not available [93,102]. Storage times for these commercial products are claimed to range from three days up to 30 days at temperatures ranging from 4 °C up to 37 °C (Appendix A). Although not claimed by the manufacturer, the LBgard Blood Tube (Biomatrica, San Diego, CA, USA) was reported to be effective in preserving cfDNA for up to 72 days. On the other hand, the majority of the collection tubes found by us are officially dedicated to the analysis of cfDNA or CTCs, while there are only two devices, the RNA Complete BCT tubes (Streck, La Vista, NE, USA) and the cf-DNA/cf-RNA Preservative Tubes (Norgen, Thorold, Ontario, Canada) reported to be suitable for the stabilization of cfRNAs in the blood (Appendix A). However, apart from the claimed utility for cfDNA, both the PAXgene Blood ccfDNA Tubes (Qiagen, Hilden, Germany) and the Cell-Free DNA Collection Tubes (Roche, Basel, Switzerland) were found to be suitable for the preservation of cfRNA too, specifically of cf-miRNAs [96].

When compared to EDTA, and considering the effectivity of cell lysis prevention, claimed utilities of the available systems were proved in several studies, including the Streck Cell-Free DNA BCT tubes [13,26,91,92,93,95,103,104], the CellSave Preservative tubes (CellSearch, Huntington Valley, PA, USA) [97,105], the Qiagen PAXgene Blood ccfDNA Tube [92,106], the Roche Cell-Free DNA Collection Tubes [12,98,107], the Norgen cf-DNA/cf-RNA Preservative Tubes [96] and Biomatrica LBgard Blood Tube [108,109,110]. Studies listing limitations are, however, also available. The suggested storage for up to 14 days was not confirmed, for example, for the Streck Cell-Free DNA BCT tubes in a study of Zhao et al. [12]. It should also be noted that temperatures outside the recommended range significantly affected the yield of cfDNA due to unprevented cell lysis, highlighting the importance to maintain temperatures above 10 °C and up to 37 °C to ensure plasma quality [13,94,111]. When using the Streck Cell-Free DNA BCT tubes, Qin et al. demonstrated their ability to stabilize proteins and mRNA for up to 4 days at room temperature [105], however, other studies claimed that they are not suitable for the parallel analysis of other biomarkers, including cfRNA, and in some cases requiring modifications of the DNA extraction procedure [13,92,96]. The Streck Cell-Free DNA BCT tubes were reported to be inappropriate for the analysis of methylated sequences of cfDNA [106], the Qiagen PAXgene Blood ccfDNA Tubes was reported to have no effect on methylation analyzes, in line with the manufacturer’s claims [106]. On the other hand, another study claimed that neither the Streck Cell-Free DNA BCT tubes nor the Qiagen PAXgene Blood ccfDNA Tubes are suitable for the analysis of methylations [112].

When considering downstream applications, several studies compared common commercial products. Sherwood et al., for example, demonstrated a significantly decreased detection rate of mutations after 72 h when stored in EDTA tubes compared to the Streck Cell-Free DNA BCT tubes and that different input of plasma volume can affect the DNA yield [91]. In another comparison, EDTA, the Streck Cell-Free DNA BCT tubes, and the CellSearch CellSave Preservative tubes were found to stabilize ctDNA at room temperature, or at 4 °C, for up to six hours before plasma processing, however, the Streck Cell-Free DNA BCT tubes and the CellSearch CellSave Preservative tubes demonstrated higher tumor mutation fraction than EDTA in which significant reduction of ctDNA occurred [94]. The Roche Cell-Free DNA Collection Tubes, the Qiagen PAXgene Blood ccfDNA Tubes, and the Streck Cell-Free DNA BCT tubes were found to be similarly effective in stabilizing ctDNA for a long time, however, in a case of low ctDNA concentration in blood only the Roche Cell-Free DNA Collection Tubes and the Qiagen PAXgene Blood ccfDNA Tubes were sufficient for the tumor mutation detection [113]. In the comparison study of Ward Gahlawat et al., the Norgen cf-DNA/cf-RNA Preservative Tubes and the Qiagen PAXgene Blood ccfDNA Tubes were found to be most suitable for the analysis of cfRNAs, while the Norgen cf-DNA/cf-RNA Preservative Tubes for combined analysis of both cfDNA and cfRNA [96]. In a study performed by Van Paemel et al., the Streck Cell-Free DNA BCT tubes, the Qiagen PAXgene Blood ccfDNA Tubes, the Roche Cell-Free DNA Collection Tubes, and Biomatrica LBgard Blood Tubes performed comparably in several downstream applications [108].

Since the introduction of the Streck Cell-Free DNA BCT tubes, various other collection tubes have been developed, while only a few of them were cross-evaluated for their effectiveness to stabilize and preserve cfNAs. There are, however, also newer collection systems, such as the CEE-Sure BCT (Biocept, San Diego, CA, USA), the Blood STASIS 21-cfDNA Blood Collection Tubes (MagBio Genomics, Gaithersburg, MD, USA) the Check cfDNA Tube (EONE-DIAGNOMICS NICE, Incheon, Korea) and the ImproGene Cell Free DNA Tube (Inresearch, Tsuen Wan, Hongkong) which were yet not studied as extensively as the abovementioned ones (for more detailed characteristics see Appendix A).

### 3.2. Saliva

Saliva represents viable body fluid, which, unlike blood, does not coagulate, therefore it is easier to collect, transport, and store. On the other hand, saliva mainly consists of water, mucins, enzymes, and proteins, which all may affect downstream analyzes, while salivary proteins, cfNAs, and cfRNAs containing exosomes may very rapidly degrade when they are out of their environment [114]. Protocols for the whole saliva and also glandular secretion collection were established [115], however, these were not focusing on the stabilization of cfNAs, neither optimized for all downstream molecular analytical methods [116]. Like blood, therefore, an optimized process for collection and processing is required for the stabilization of the salivary components including inhibitors for preserving the integrity of analyzed cfNAs or exosomes. Saliva can be collected under two conditions, —stimulated or unstimulated manner—by methods including spitting, suction, swabbing, and passive drooling. Preparation of salivary supernatant, encompassing collection on ice, centrifugation at 4 °C and addition of RNase and protein inhibitors, although labor-intensive, were found to be effective in improving cfRNAs and proteins stability [117]. Since saliva represents a valuable body fluid useful in early diagnostics, commercially available devices for the saliva collection were also developed. However, these are mainly designed for the analysis of genomic DNA, RNA, or proteins (Salivette from Sarstedt (Numbrecht, Germany), Saliva Collection System from Greiner Bio-One (Kremsmunster, Austria), SalivaBio Oral Swab from Salimetrics (Carlsbad, CA, USA), etc.), but some of them may also enable analyzes of cfNAs and exosomes [118]. On the other hand, when compared to blood, there is a significantly less heterogeneous collection of available commercial products for salivary cfNAs sampling (Appendix A), while the Pure•SAL and RNAPro•SAL (Oasis, Vancouver, WA, USA) were reported to be suitable for the analysis of salivary cfNAs [117], the Saliva Exosome Collection and Preservation Kit (Norgen, Thorold, ON, Canada) was proved to be useful for cfRNAs isolation and downstream analyzes [119]. According to the manufacturer of this latter kit, salivary exosomes in preserved samples are stable for two years at room temperature (Appendix A), however, we yet did not find an independent evaluation of these claims.

Saliva is viscous and may consist of food residue or other particles, therefore its pretreatment is necessary. However, not all methods are suitable when analyzing cfNAs or exosomes. For example, filtration methods aim to reduce viscosity by removing glycoproteins, but since exosomes are larger than proteins, they can be lost [30]. Several studies have been performed regarding the cfNAs analysis from saliva, however, only a few reported the exact collection tube or stabilizer agent used for the saliva collection [31,120,121,122] as an example we present a study in which the Carlson Crittenden parotid collectors were used [123]. However, these are specifically developed collection devices for the type of the gland and not for the stabilization of the exosomes or cfNAs. Saliva represents a promising body fluid, therefore studies are focusing not only on the cfNAs analysis but also on the analysis of the salivary proteome. In these studies, saliva collection and processing were mainly performed either in accordance with previously published protocols (rinsed out the mouth with water and spit into 50 mL sterile Falcon tube) without adding any stabilizer [31,121,122,124] or collected into Falcon tubes and adding a stabilizer [125]. In other studies, the collection and processing were performed according to Navazesh and Christensen [126] or Navazesh protocols [127] with adding RNase and protease inhibitors [120,128,129] or using dedicated collection devices and immediately after saliva collection protease inhibitors were added [130]. According to our knowledge, only one study evaluates different conditions of saliva samples storage. Its results have shown that salivary proteomes could be stabilized if the samples are kept at 4 °C with protease inhibitors or by adding absolute ethanol to the samples kept at room temperature [125]. Although saliva seems to be the easiest body fluid for the analysis of cfNAs as a liquid biopsy, several factors should be considered in saliva collection. Different glands produce different levels of salivary analytes [131], therefore the location of sampling is very important. Saliva composition varies daily and is affected by many external factors, such as food, drinking, smoking, brushing the teeth, exercise, kissing, etc. [132,133], therefore the time and instruction for proper collection are also very important.

### 3.3. Urine

Although urine represents a highly promising body fluid, yet little is known about the urinary cfNAs and the methods for their processing. Urinary cfDNA has a shorter half-life than blood derived cfDNA [134] because of high DNase I [135] and DNase II activity [37]. Therefore, inhibition of DNases might be crucial to prevent cfDNA degradation. One of the effective ways is to use treatment with EDTA and store urine samples at −70 °C or below for a long-time or to use sterile containers and store samples at 4 °C or on ice and process them as soon as possible [136]. Despite such conditions, the loss of cfDNA may occur [137]. Acquisition of high yields of cfNAs and their subsequent sensitive detection as biomarkers depend, therefore, on appropriate preanalytical conditions. When trying to evaluate these processes in the literature, we found that some of the studies do not describe the urine collection devices nor any preservative agent [138,139,140,141,142], while others reported to use a sterile container without any stabilizer [121,143], used EDTA as a stabilizing agent [144,145], or used commercial preservation solution [146]. Only a few studies, to our knowledge, focused on the optimization and standardization of the preanalytical process for urine cfNAs studies.

Various commercial stabilization and preservation solutions were developed and used for cfNAs applications (Appendix A), including the Streck Cell-Free DNA Urine Preserve [147], Urine Conditioning Buffer [148], Norgen Urine Preservation [149,150], Colli Pee (DNA Genotek, Ottawa, ON, Canada) [151], or the Urine Collection Tube (Hunan UPSBio, Hunan, China) [136]. They are generally liquid reagents which can be added to the urine samples, or they may be available in a dried form in the collection tubes, to stabilize cfDNA and inhibit nuclease activity. Manufacturers claim storage times in a range from days up to years and storage temperatures mainly around room temperature (Appendix A). Although the suitable temperature ranges from 6 °C to 37 °C for the Streck Cell-Free DNA Urine Preserve, samples with this commercial preservative kept at 4 °C were found to be less stable, when mimicking transport condition, suggesting that storing or transporting samples at room temperature, or at higher temperatures, might be more suitable [147]. Unfortunately, studies evaluating the efficacy of these commercial products are largely missing. Recently, Augustus et al. compared the influence of the different preanalytical conditions on the concentration of cfDNA and gDNA. They found that it is inadequate to store samples without any preservatives for a week at room temperature resulting in depletion of a majority of DNA, however, keeping them at 4 °C can slow down the degradation and samples may remain stable for up to 96 h. Although their in-house developed preservative was shown to effectively preserve total DNA in urine, the Streck Cell-Free DNA Urine Preserve led to better results in the preservation of cfDNA [147].

Important differences in preanalytical conditions may exist, however, when evaluating DNA and RNA stability. When considering miRNA content encapsulated in EMVs, for example, the optimal handling condition includes sample storage at room temperature (around 20 °C) and further processing as soon as possible after collection. Alternately, urine held at 2 °C to 4 °C and processed within 24 h, as well as one-time freeze-thaw samples, may be used for miRNA profiling, however, a 5 min 37 °C sample equilibration to solubilize any precipitation for increased miRNA recovery is recommended. On the other hand, it is important to note that urine samples held at 37 °C for as little as four hours are significantly compromised and are not suitable for miRNA profiling [152].

Similarly to saliva, urine seems to be easy body fluid for liquid biopsy, however, amounts of cfNAs vary daily, may be higher in females than in males [153] and first morning urine seems to contain more degraded cfDNA [144]. It is essential to use freshly voided urine to limit the potential bacterial presence and also lifestyle or dietary changes should be considered [154]. All of the above points account to the numerous factors which could impact the downstream analyzes, so they should be kept in mind when designing analytical processes.

### 3.4. Collection and Stabilization of Less Commonly Analyzed Body Fluids

Less conventional body fluids, for which standardized or commercial collection systems are yet missing, are generally collected into systems originally dedicated to other body fluids, or for the same body fluid but different analyte, such as conventional blood tubes or sterile containers [64], supplied with EDTA or even without any stabilizer. In such cases, however, the performance of the system should be evaluated, or the samples should be processed as soon as possible.

Since stool contains a really large portion of other components that could inhibit downstream analyzes, it is necessary to choose a collection method which allows the maximal recovery of human DNA, preserves human NAs, and inhibits undesirable components. There are some commercial collection devices or storage systems, containing specialized preservatives for the stabilization and protection of NAs in stool samples, available, including the DNA/RNA Shield Fecal Collection tubes (Zymo, Irvine, CA, USA) [155], the Norgen Stool Nucleic Acid Collection and Preservation Tubes [156,157], or the OMNIgene-GUT buffer (DNA Genotek, Ottawa, ON, Canada) [158], however, neither of them were found to be dedicated to cfNAs. They generally prevent the growth of many bacteria, fungi, and inactivate viruses, and also offer stabilization of NAs in stool samples from several days up to a few years, most commonly at room temperature, or in a frozen state (Appendix A). However, it is not yet clear, whether these stabilization solutions are effective for the preservation of the wide range of different DNA length fragments [158].

For the collection of sputum, for example, there are various collection methods developed, likely due to the general use of the sputum in the detection of *Mycobacterium tuberculosis*. For cfDNA analyzes samples were reported to be collected into Fixcyt, comprising ethanol, polyethylene glycol, and ultrapure water [61], or into a mixed solution of Saccomanno’s fixative and dithiothreitol [62]. However, not each study describes the collection method in detail [159]. Breast milk collected into centrifuge tubes without any stabilizer was reported to contain relatively stable cfNAs even at room temperature for a week [63]. For milk samples, on the other hand, Norgen Milk DNA Preservation and Isolation Kit offers DNA preservation for up to one month at room temperature [160], however, it is not specified whether this kit preserves also cfDNA. CSF was reported to be collected, following lumbar puncture, into the Streck Cell-Free DNA BCT tubes to prevent the release of gDNA [49], but there are also studies which do not provide details regarding the collection procedure [48,50]. Ejaculate was reported to be collected in sterile containers [56], then was allowed to liquefy within 30 min at room temperature or at 37 °C, while in some studies a cryopreservation was added [55], however, there are also studies which do not mention adding any stabilization nor describe the whole collection step [53,54,57].

### 3.5. Concluding Remarks for Sample Collection Possibilities

Lampignano et al. suggested that downstream analytical processes, such as massively parallel sequencing (MPS) and droplet digital PCR, could be utilized with a variety of collection tubes [161], however, there are some exceptions, usually when methylation analyzes should be performed. For example the Streck Cell-Free DNA BCT tubes do not seem to work for such analyzes, therefore the use of the Qiagen PAXgene Blood ccfDNA Tubes should be considered when designing methylation assays [106], however, there are also reports of the use of the Streck Cell-Free DNA BCT tubes for epigenetic analysis of cfDNA [162]. Although K_2_EDTA showed worse effect then the Streck Cell-Free DNA BCT tubes on the cfDNA and ctDNA recovery, it seems that on the quality of MPS analyzes neither of these preservatives have noticeable effect [26], even the Streck Cell-Free DNA BCT tubes can be used for the library preparation for aneuploidy detection in prenatal care without any unwanted changes [93]. These tubes proved their suitability for PCR and real time quantitative PCR [101], but also for less conventional methods (e.g., BEAMing and Plasma Safe-Sequencing technologies) [13]. It was demonstrated that the Roche Cell-Free DNA Collection Tubes, similar to the Streck Cell-Free DNA BCT tubes, do not affect the quantity and complexity of MPS libraries. These two factors are very important in determining the quality of MPS, since they can reflect the bias of MPS-based results caused by cfDNA quality [12]. Using a microarray technology approximately 3000 different mRNAs were discovered in cell-free saliva. It points to that not only from blood, but cfNAs originating from all body fluids should be acquired in sufficient quality and quantity for plenty of downstream analysis regardless of whether they are performed in a diagnostic manner or in research [162,163].

## 4. Processing and Storage of Biological Material

It is important to note that NAs extracted from body fluids are not automatically cell-free or extracellular, since the majority of body fluids contain also cells, either being physiologically present, or released into the fluid from the surrounding tissues, or even representing the microbiome. These need to be removed, most commonly by a series of centrifugation and/or filtration steps. Since these cells may release their cellular NAs content following lysis, the faster the processing is, the less contamination by cellular NAs may occur. Although the procedures may be different, there are also examples when the same processing procedures were used for several body fluids, like in the study of Weber et al., in which 12 different body fluids were studied and processed under the same conditions [60]. In addition, there are also procedures which allow to differentially isolate EMVs present in body fluids. Some of the extra steps used for exosome separation are described in the following text, however, there is also a variety of commercial kits dedicated to exosome isolation, which are suitable also for the isolation of other EMVs. These are summarized in Appendix A and are most commonly based on immunoassays, on size exclusion chromatography, or on precipitation, along with different other possibilities. Alternative approaches for isolation of exosomes may utilize microfluidics, which may employ methods such as immunoaffinity [164,165,166], ciliated micropillars [167], porous polymer monoliths (PPM)-based membrane filters [168], functionalized surfaces [168,169], and droplet-based systems [170,171].

### 4.1. Plasma Preparation

Numerous centrifugation protocols were developed for blood plasma acquisition. Centrifugation of whole blood allows to separate liquid (serum and plasma) and cellular fraction, representing an important step to prevent the plasma contamination by cellular NAs. Usually, plasma preparation protocols are based on different centrifugal forces (lower or higher forces, or even ultracentrifugation), steps (one-step or two-step protocols) and temperatures (4 °C or at room temperature). Lower cellular DNA contamination was reported when the blood was processed by double centrifugation. It was found that double centrifugation at 2000× *g* for 10 min did not decrease DNA yield relative to single centrifugation of blood processed two hours post-collection, but the second step significantly reduced DNA yield in blood processed 72 h post-collection, suggesting the presence of contaminating cellular DNA at 72 h due to lysis of blood cells that was not efficiently pelleted by a single centrifugation step [13,91]. Another described possibility is the use of an initial slow-speed centrifugation at 1600× *g* for 10 min followed by a second centrifugation of the collected plasma at high-speed 16,000× *g* for 10 min [91,172]. The same two-step protocol with identical centrifugal forces but with different time and temperature, i.e., 15 min at 22 °C, was also reported [173], while double centrifugation protocols were reported to be performed both at 4 °C [12,91,174,175] as well as at room temperature [176]. In each case the samples should be removed carefully after the first centrifugation to avoid contamination by the buffy coat during the plasma separation [177]. It has been observed that combination of higher speed centrifugation at 14,000× *g* with the following filtration by 0.45 μm filter is suitable to isolate the purest plasma fraction when compared to low-speed centrifugation at 1600× *g* [178,179], however, no centrifugation time was specified in this study. A third centrifugation step at 16,000× *g*, on the other hand, did not result in any difference in the yield of plasmatic cfDNA [178]. It should be noted that there are also studies which did not observe statistically significant changes in plasma cfDNA yields using different centrifugal forces [95,180].

The plasma storage conditions represent critical issues too, influencing, for example, cfNAs fragmentation. Once plasma is separated, it should be stored frozen at −20 °C [181] or even at −80 °C for a long time storage [182], while due to cfDNA fragmentation no freeze-thaw cycles are recommended [183]. Plasma storage for two weeks at −80 °C and for four weeks at −20 °C was reported to have no effect on cfDNA extraction [184]. Longer periods of storage at −80 °C, for example 5–21 months, may, however, cause a decreased cfDNA yield [184,185]. On the other hand, plasma samples stored at −80 °C from one to six years were reported to be still suitable for analyzes using MPS [186]. However, different types of collection tubes, together with external factors such as extreme temperature or long-time storage of the biological material in the collection tubes also affect processing. Under certain circumstances almost no visible clear plasma was present after the first centrifugation, for example in the Streck Cell-Free DNA BCT tubes [13]. Regardless of the collection tubes, volume of separable plasma was found to decrease over time by storage at room temperature [26].

Process of plasma preparation for cfRNAs extraction may be little different from that for cfDNA extraction. To separate plasma, for example, samples may be processed by two-step centrifugation too, but at 300× *g* for 20 min at room temperature at first, and after removal of the upper plasma layer and transferring to a new tube, the upper plasma may be centrifuged at 5000× *g* for 10 min, followed by storage at −80 °C [103]. The study of Sorber et al. provides a good comparison of different centrifugal forces for high-quality cfDNA and cfRNA yield in both EDTA and the Streck Cell-Free DNA BCT tubes [103,180], while the study of Zheng et al. compared one-step and two-step centrifugal methods that are routinely used in plasma miRNA analysis [187]. Since most miRNA molecules are bound to proteins [188] or encapsulated in vesicles, it has been shown that double centrifugation protocol decreases the quantity of plasma miRNAs, because exosomal or protein complexes containing miRNAs may be precipitated in a certain degree after two-step centrifugation [187]. In line with this, plasma preparation by the two-step centrifugation significantly resulted also in lower cf-miRNA levels, when compared to both plasma and serum samples processed by low-speed centrifugation [189]. In case of cf-lncRNAs analyzes, single low-speed centrifugation at 800× *g* for 10 min at room temperature can be used [190].

In addition to cfDNA and cfRNAs, cf-mtDNA can also be isolated from pure plasma samples. When comparing four different blood-processing treatments, including a standard double centrifugation protocol, filtration with 0.22 μm filter, ultracentrifugation at 99,960× *g* for two hours and no additional treatment, the two-step centrifugation followed by filtration or ultracentrifugation is more efficient for removing particle-associated mtDNA [191]. When combined with the recently described protocol for total RNA sequencing, which allows assembly of the complete mitochondrial genome, it could be an effective method for cell-free whole mitochondrial genome analysis [192].

After the cell-free plasma preparation and removal of cellular components, exosomes may be isolated using size exclusion chromatography [193], using differential centrifugation coupled with ultracentrifugation [194], epithelial cell adhesion molecule immunoaffinity pull-down, or using OptiPrep density gradient separation, while when comparing the latter three methods, OptiPrep density gradient was found to be superior in isolating pure exosomal populations from blood plasma [195].

### 4.2. Saliva

The various potential biomarkers and the genetic material within saliva may require different conditions for optimal preservation. Furthermore, storage methods may differ based on the type of downstream application needed. Samples should be processed as soon as possible after collection, ideally within 24 h or stored immediately under ultralow temperatures at −80 °C. During transport, samples should be kept at 4 °C. A study by the UK Biobank showed that there was a little change in the quantity of many salivary fluid components, including genetic material such as mRNA, under the Biobank’s normal handling conditions, which involves keeping saliva samples at 4 °C for 24 h prior to freezing them at −80 °C [196]. Moreover, the quantity and integrity of DNA and RNA within whole saliva samples have been shown to be relatively preserved for at least five years when samples were kept at −80 °C [197]. To remove cell debris, bacterial cells and contaminants, whole saliva samples require centrifugation. For cfDNA isolation, for a maximum of one hour after collection, the saliva samples are centrifuged at 300× *g* for 20 min and followed by at 10,000× *g* for 20 min [31]. In the study of El-Mogy et al. [198] the saliva samples were processed by single-step centrifugation at 200× *g* for 10 min to isolate small cfRNAs. Centrifugation at 3000× *g* for 15 min at 4 °C was used by Yu et al. to exosomal *PD-L1* mRNA isolation [199].

As the gold standard method for salivary exosome isolation density gradient or sucrose cushion ultracentrifugation at 100,000× *g* was reported [200]. On the other hand, ultracentrifugation protocols at 160,000× *g* for one hour at 4 °C for exosomal isolation were used with previous double centrifugation at 1500× *g* for 10 min and 17,000× *g* for 15 min at 4 °C to remove cell debris [123]. In recent years, several exosome and EMVs extraction protocols have been proposed and some of them were summarized by Konoschenko et al. [201]. For example, to remove contaminants, the saliva samples could be centrifuged at 3000× *g* for 15 min. Then the supernatant is filtered with 0.2 μm filters and precipitation solution was added. Overnight incubation at 4 °C with following centrifugation at 3000× *g* for 30 min to vesicles precipitation and 1500× *g* for 5 min to remove residual supernatant were performed [122]. For exosomal precipitation the refrigeration at 4 °C overnight was also used, followed by exosomes collection using a two-step centrifugation at 1500× *g* for 30 min [199].

### 4.3. Urine

Centrifugation of the whole urine sample is required to separate the cell-free supernatant from residual cells. This could be performed as a single high-speed centrifugation, for example at 16,000× *g* for 10 min [202], or a single low-speed centrifugation, for example at 850× *g* [148], that was reported to be enough to well separate cells with no change in the yield and the quality of urinary cfDNA. Other single step protocols were also reported to be functional, such as single centrifugation at 4000× *g* for 10 min, or 3000× *g* for 15 min [147], suggesting a wide functional range for centrifugation protocols. On the other hand, there are studies that prefer a double centrifugation protocol, first at 200× *g* followed by 1800× *g* for 10 min [153], or first low-speed centrifugation at 1600× *g* for 10 min followed by second high-speed centrifugation at 16,000× *g* for 10 min at 4 °C [136]. Two-step centrifugation at 200× *g* for 10 min followed by a second centrifugation at 1000× *g* for 10 min was also reported to be used [152]. After removal of any residual contaminants, the cell-free supernatants are usually stored at −80 °C until further analysis [152,203,204].

Similarly to other body fluids, exosomes can be isolated from urine samples using a combination of several methods, including differential centrifugation coupled with ultracentrifugation, ultrafiltration, precipitation with polyethylene glycol, size exclusion chromatography and immunoisolation, which are summarized by Zhang et al. [205]. Although ultracentrifugation is a widely used method, combination of size exclusion chromatography and ultrafiltration may provide even higher urinary exosome concentration [206]. One of the alternative approaches to urine exosome isolation is filtration. There are many different methods based on filtration, including nanomembrane concentrators with a membrane pore size small enough to block exosomes and large enough for water, electrolytes, and soluble proteins to pass through. Centrifugation is used to enhance flow, but the forces required are lower, unlike ultracentrifugation [207]. Another option is to use dialysis membranes, although the purification time is typically longer [208]. Alternatively, size exclusion chromatography with exosomes collected in the initial flow-through fraction may be used [209]. Exosomes can be also precipitated by altering the properties of the surrounding fluid and further isolated by centrifugation at lower speed using standard laboratory centrifuges [210].

There are also commercial reagents available based on volume-excluding polymers [211] and several different polymers, while different molecular weights and concentrations of such polymers are suitable for certain body fluids. Polymers are sequestering water causing the precipitation of less soluble material. This method is simple, but the purity of the preparation is questionable [212,213]. Exosomes can also be isolated based on their affinity to different substrates, such as heparin [214] and lectins [215] which have been used for their agglutination. Spin column-based kits [216] are now available utilizing an electrostatic interaction between the negatively charged exosomes and a capture surface functionalized with positively charged ions [217]. The greatest specificity offers the use of antibody-based affinity. This method potentially enables capture of exosomes from specific cell types or simultaneous capture and detection [218]. Exosomes can be further isolated using antibodies conjugated to beads [219], magnetic beads [220], porous silica substrates [221], and directly in coated assay wells [218]. After urinary exosomal isolation the samples should be stored at −80 °C until further analysis [222].

### 4.4. Processing of Less Commonly Analyzed Body Fluids

When compared to conventionally used body fluids, standardized operational procedures for the processing of less commonly used biological fluids for cfNAs analyses is even scarcer. In case of CSF, the processing of collected samples was reported to involve either a one-step centrifugation at 1000× *g* for 10 min at 4 °C [223], a double-step protocol at 1500× *g* for 10 min at 4 °C and 20,000× *g* for 10 min, with subsequent long-term storage at −80 °C [48], or a single centrifugation at 2000× *g* for 10 min at 4 °C for cf-mtDNA isolation [224].

Bronchoalveolar lavage fluid samples were processed by one-step centrifugation at 2000× *g* for 30 min at 4 °C [225], at 350× *g* for 10 min at room temperature [226], or at 1000× *g* for 10 min at 4 °C [72].

According to the WHO guidelines, to prevent sperm lysis, samples should be processed within two hours from collection at low-speed centrifugation 400× *g* for 10 min at room temperature, followed by high-speed centrifugation of the seminal plasma containing supernatant 16,000× *g* for 5 min at room temperature. Seminal plasma samples should be stored at −80 °C until cfDNA extraction [227,228]. For isolation of exosomes from seminal plasma, after liquefication at 37 °C for 30 min, 1600× *g* for 10 min for first centrifugation and 16,000× *g* for 10 min at 4 °C for second centrifugation was used that was followed by one microfiltration step, with a pore size 0.22 μm, and ultracentrifugation at 100,000× *g* for two hours at 4 °C [229,230]. Another study suggested, however, that seminal plasma filtered through the 0.22 μm filter resulted in similar cfDNA yield and size distribution than seminal plasma acquired by high-speed centrifugation (16000× *g* for 5 min) with or without initial low-speed centrifugation (initial spin at 400× *g* for 10 min followed by seminal plasma spin at 2000× *g* for 20 min) [54].

For the processing of whole breast milk a single-step centrifugation protocol was reported to be used, involving 1000× *g* for 10 min for RNA extraction [60], or a two-step protocol, involving centrifugation at 1500× *g* for 10 min and at 10,000× *g* for 10 min at room temperature for cfDNA and cfRNA extraction [63].

Based on the few studies published, after removing the cells and debris by centrifugation, the pleural fluid supernatant should be used for cfNAs testing, while the described centrifugation protocols are mostly one-step, involving 1600× *g* for 15 min [231], or 1000× *g* for 10 min at 4 °C [232], with subsequent storage of supernatant either at −20 °C, or at −80 °C, respectively.

In one study the same centrifugation protocol was used for the preparation of the cell-free ascites and PE supernatants. After the centrifugation at 1500 RPM (listed without specification of the rotor radius or centrifugal forces) for 5 min the samples were stored at −80 °C [71].

Like most other body fluids, pancreatic juice samples were reported to be processed for exosome isolation and cf-miRNA extraction by a two-step protocol combined with filtration and ultracentrifugation. Exosomes could be extracted by ultracentrifugation according to isolation protocol summarized in the study of Costa-Silva et al. [233]. Next, the pancreatic juice was spun at 300× *g* for 10 min at 4 °C followed by next centrifugation at 16,500× *g* for 20 min at the same temperature. Before ultracentrifugation at 140,000× *g* for 70 min at 4 °C, the supernatant was filtered through a 0.2 μm filter [234].

For tear fluid, after the sample collection a brief spin in a microcentrifuge is recommended, and then the tear samples should be kept on ice until cfDNA analyzes [59]. Compared to cfDNA, isolation of EMVs from tears includes initial centrifugation step at 20,000× *g* for 15 min at room temperature followed by dilution and filtration through 100 nm pore-size syringe filters and ultracentrifugation of filtrate at 100,000× *g* for 90 min at 4 °C to obtain a pellet containing EMVs [235].

For the processing of frozen amniotic fluid samples, thawing at 37 °C, vortexing for 15 s and centrifugation at 13,500× *g* was reported [67].

Although there are several studies describing cfNAs analyzes from less conventional body fluids, methodological sections of many of them contain incomplete information on analytical details, with the temperature at which samples should be processed missing relatively commonly. This pre-analytical factor, as well as the other steps, may influence the quantity and quality of isolated cfNAs. However, most studies agree that a second centrifugation step of suitable force reduces the cells and cell debris that can contaminate samples by cellular DNA to a minimum.

## 5. Approaches for the Extraction of cfNAs from Various Body Fluids

Although laboratory developed assays, such as the conventional ethanol precipitation, phenol-chloroform or triton-heat-phenol-based assays, may be used for several cfNAs extraction procedures, the present day market is well equipped with commercial kits dedicated to the isolation of cfDNA and cfRNAs from the most commonly used body fluids, mostly using silica-based spin columns or magnetic beads [236], or even the so called polymer mediated enrichment (PME) technology, which capture cfDNA by encapsulating it within the PME polymer (Appendix A). In contrast to the previous processing steps, the “most commonly used” category in this section does not contain saliva, since we did not find commercial extraction assays dedicated to cfNAs for this body fluid.

### 5.1. Extraction of cfNAs from Blood Plasma or Serum

The most commonly used body fluid is blood, more specifically plasma, and since the prepared plasma is usually frozen at −80 °C, samples should be thawed prior to the extraction step and immediately processed using an appropriate cfNAs extraction kit [13]. However, these recommendations may be modified based on the recommendations of the manufacturer of the collection tube or preservation solution used for sample collection and stabilization. Despite dedicated efforts during sample processing, cfNAs in the blood may still face contamination from cellular NAs released after sampling, strongly suggesting the benefits of using extraction kits developed for cfNAs, however, the use of different kits may provide different yields of cfNAs. It should be mentioned, on the other hand, that extraction kits developed for other purposes, such as for isolation of viral NAs, were also widely used for cfNAs applications [237].

When concerning cfDNA extraction kits, in many applications the commercial spin column kits containing silica membranes are widely used [91,238,239], in which NAs bind to the silica surface under high chaotropic salt conditions and are detached at low salt concentration [240], providing relatively high yield and high purity of cfDNA [241]. From these, several comparative studies have identified either the Qiagen QIAamp Circulating Nucleic Acid Kit as one of the best-performing commercial options for plasma cfDNA extraction [238,242,243,244,245], or the Norgen Plasma/Serum Cell-Free Circulating DNA Purification Kit [239]. With regard to this latter, although the kit has high overall yields from plasma, its recovery has been previously shown to decrease with the length of fragments [238,244].

Raymond et al. described a two-step cfDNA purification method, relying on magnetic bead-based technology that entails extraction of total DNA followed by size-selective enrichment of the smaller fragments that are characteristic for DNA originating from fragmentation between nucleosomes. The first step of this method is extraction of total DNA from plasma with an emphasis on near-complete recovery of DNA present in the sample. The second step is size-based separation of nucleosomal-sized cfDNA fragments from high molecular weight gDNA, while the yields of cfDNA were reported to be directly comparable to the most popular QIAamp kits, but with lower overall costs [246].

The alternative approaches for the isolation of cfDNA are microfluidic and lab-on-a-chip technologies that can automate the process and reduce the time and labor cost. Compared to the common isolation kits, microfluidic technologies have several advantages, such as high throughput, sensitivity, purity and lower operating costs. There are various solid phase microfluidic extraction methods based on an extended surface area in microchannels [247,248,249], miniaturized fluidic chips including silica membranes [250,251], or silica beads [252,253,254]. For cfDNA extraction and a cancer monitoring test a lab-on-a-disc system containing silica-coated beads [255] and surface modification using a non-chaotropic agent, dimethyl dithiobispropionimidate (DTBP) were used [256]. However, there are several aspects of the system, which need to be further developed for the use in clinical settings. These aspects include full automatization, large working capacity for simultaneous multiple extraction in a single operation, complete incorporation of separate experimental steps, and ease of use with low total cost. Lee et al. introduced a pressure and immiscibility-based extraction (PIBEX) method for the centrifugation-free extraction of cfDNA with a silica membrane under vacuum pressure using an immiscible liquid, such as mineral oil. However, microfluidic integration and automatic fluid control were not realized [257]. Principle of the PIBEX is based on the microfluidics associated with surface tension in silica membrane pores. Owing to these methods the entire extraction within a chip can be replaced with one continuously operating process.

### 5.2. Extraction of cfNAs from Urine

For the extraction of cfDNA from urine commercial kits based on silica gel membrane or magnetic beads can be used [137,145,147,153]. Comparison studies are also available for several of them and in one of them the MagMAX Cell-Free DNA Isolation Kit (Thermo Fisher, Waltham, MA, USA) and the Qiagen QIAamp Circulating Nucleic Acid Kit provided the highest cfDNA yield in the 50–300 bp range, while the Thermo Fisher MagMAX Cell-Free DNA Isolation Kit and the Norgen Urine Cell-Free Circulating DNA Purification Midi Kit gave the highest cfDNA yield in the 50–100 bp range. In particular, the Urine Cell-Free Circulating DNA Purification Midi Kit was efficient for the isolation of more fragmented cfDNA in the range of 50–100 bp with the lowest cellular gDNA contamination. Another study reported that the Zymo Quick-DNA Urine Kit had the best cost-efficiency for isolating the same amount of urinary cfDNA [137].

However, it is possible to use other extraction methods too, such as poly-Lys-coated particles, which demonstrate strong affinity for negatively charged molecules such as DNA. The amounts of cfDNA (1.1–1.5 μg) recovered using this method was reported to be about three times higher than that previously obtained using the commercial Zymo Quick-DNA Urine Kit, with high purity [258]. However, fragment sizes of about 20 bp, while other studies reported size ranges of cfDNAs around 100–200 bp in urine [17,138,258,259,260,261], may suggest lower extraction efficiency for long DNA fragments as compared to that for short ones [258]. In a modified version using amine-modified silica particles, surface area and modification significantly improved DNA extraction [262].

The Wizard resin/guanidinium thiocyanate (Wizard/GuSCN) method uses high concentrations of chaotropic GuSCN to adsorb DNA to Wizard silica resin, but was reported to have low and variable recovery, with improved recovery of moderately short targets of about 40 bp, and inability to recover the shortest fragments of about 25 bp [145]. The Q Sepharose method uses a quaternary ammonium anion exchange resin to preconcentrate DNA before desalting on a silica spin column, while is characterized by a user friendly, ready-to-go protocol [145,263]. When extreme sensitivity and retention of the shortest fragments are required, use of hybridization capture is recommended instead [145].

Laboratory developed extraction method based on hybridization capture for urine cfDNA using a biotinylated sequence specific probe and streptavidin-coated magnetic beads has perspective for urine cfDNA applications where maximum sensitivity is required, especially in the case of an ultrashort PCR target, (e.g., 25 bp). A key limitation of hybridization capture is that, unlike silica-based methods, it will only isolate specific targeted sequences [145].

The Norgen Urine Cell-Free Circulating DNA Purification Kit uses a hybrid silica/silicon carbide spin column, where addition of silicon carbide reportedly improves yield of short DNA compared with silica alone [264]. The biggest disadvantage of this method is consistent PCR inhibition, even when using only a small volume of eluate in PCR. Therefore, this kit is unreliable for quantification [265]. Qiagen QIAamp Circulating Nucleic Acid Kit uses a silica vacuum column and reportedly improves recovery of fragmented DNA compared with other Qiagen kits. It is one of the most widely used commercial kits for plasma cfDNA extraction [266] but is not commonly used for urine cfDNA. The results suggest that the QIAamp kit is inadequate for capturing short DNA fragments. The Thermo Fisher Scientific MagMAX Cell-Free DNA Isolation Kit uses Dynabeads MyOne Silane to maximize binding kinetics and capacity but is intended primarily for plasma cfDNA. The use of the MagMax kit in urine samples, where most cfDNA fragments are too short, is not recommended. This is because the kit is extremely dependent on fragment length, as expected for a silica-based method.

### 5.3. Extraction of cfNAs from Other Body Fluids

Similarly to other processing steps, commercial kits for extraction of cfNAs from less commonly used body fluids are yet missing. However, since biological material coming from different body fluids tends to lose certain differences following the processing steps described in the previous section, commercial assays dedicated to other body fluids are commonly used in the literature, such as those listed for breast milk, CSF or amniotic fluid (Appendix A). Weber et al. used, for example, the same commercial miRNA extraction kit, i.e., the miRNeasy kit manufactured by Qiagen, for 12 different body fluids [60]. In other cases, like in the case of saliva, kits developed for total DNA or RNA isolation from salivary samples are reported to be used in the literature (Appendix A).

### 5.4. Concluding Remarks for Extraction Methods of cfNAs

Based on a recent study, downstream analytical processes, such as MPS and droplet digital PCR, could be utilized with a variety of extraction kits or collection tubes, however, the amount of input DNA should be considered in the process of validation [161]. It was also reported that concentration of extracted cfDNA using vacuum concentration may enhance the detection of fetal fragments from maternal plasma. On the other hand, centrifugal filters and spin columns may also be used, for example if automation is considered, for which vacuum concentration is less suitable because of an uneven and unpredictable sample evaporation rate [267].

## 6. Analytical Methods Most Commonly Used to Characterize cfNAs

Since cfNAs may have a wide range of utilities, aims of the analyzes may also vary, posing thus specific challenges for analytical methods. In applications in which the amounts of cfNAs are of major interest, simple quantification may represent the final analysis of cfNAs, while it is generally sequence nonspecific. Like in other NAs, for simple quantification of cfNAs there are at least two commonly used approaches, based on spectrophotometry and on fluorimetry, both working on different principles and having different advantages and disadvantages. In cases where the extent of degradation, contamination, or the likely origin of the fragments are also of interest, quantification may be complemented by the estimation of the size distribution of cfNAs fragments. This exploit most typically capillary electrophoresis using fluorescently stained NAs, with the possibilities of determining also the concentration of fragments representing a certain choosable size range. To extend the informational value of cfNAs, more specialized sequence specific analyses may be performed, for example for detection of the presence/absence of certain sequences using PCR amplification followed by fragment analysis on genetic analyzers, or for absolute/relative quantification using methods such as droplet digital PCR or real-time PCR, both representing quantitative PCR methods, however, working on completely different principles. From the actually available methods, however, MPS offers the most sophisticated sequence specific analytical methods for cfNAs applications, allowing, for example, relative quantification, size distribution estimations, as well as sequence and epigenetic characterization of the analyzed fragments. Importantly, sequence specific techniques allow for the detection and characterization of sequence variability, having immense biomedical significance. When considering hierarchy between these methods, it may be important to note that accurate quantification and size range estimation of cfNAs, for example using simple fluorimetry, capillary electrophoresis, real-time PCR, or even droplet digital PCR, may represent quality control steps for more sophisticated analyzes, such as MPS.

### 6.1. Sequence Nonspecific Methods Allowing Simple Quantification of cfNAs

*Spectrophotometry*, historically the first NAs quantification method, measures how much a chemical substance absorbs light by measuring the intensity of different wavelengths of the light passing through a sample solution. Measurement of absorbed or transmitted light can be used to measure the amount of a known chemical substance considering each chemical compound absorbs or transmits light over a particular wavelength range, while not only NAs have specific absorbance ranges, but also possible contaminants of the sample, such as chemicals or proteins. This can be considered for a drawback, since different substances may have the same or similar absorbance spectra, possibly leading to signal overlap and concentration overestimation [268], but also for an advantage, with an ability to provide purity measurements of the analyzed samples. Contaminants may represent spectrophotometry-associated quantification problems in a concentration dependent manner, posing thus certain complications in cfNAs applications. Therefore, despite the generally wide dynamic range of measurable concentrations (usable concentration range, for example in NanoDrop is 0.4–15,000 ng/μL, according to the manufacturer [269]), low cfNAs concentrations, together with residual organic solvents left behind after extraction, may create certain lower limits of quantification [270]. With regard to sample volumes required for analysis, spectrophotometric applications offer as small as 0.5 μL as an analyzable sample volume, when using, for example, the pedestal forms of the NanoDrop series microvolume spectrophotometers. From the viewpoint of user friendliness, there are also possibilities of analytical parallelization both in smaller scale, up to eight samples with NanoDrop, but also in larger scale, such as when using microplate readers accommodating 96- or 384-well microplates, like the SpectraMax (Molecular Devices) or Safire (Tecan) series readers. When considering simplicity, together with the time and the cost of the analyses, spectrophotometric measurements are cheaper and faster compared to other quantification methods [271].

Methods based on *fluorometry* quantification rely on the detection of an excitation driven fluorescence emission of NAs binding molecules, fluorescent dyes, which may be specific to DNA (ssDNA or dsDNA) or RNA [272]. Although they generally emit light even in an unbound state, their fluorescence emission tends to increase several folds when bound directly to their target molecules [273], so detection and quantification of NAs, including cfNAs, is highly precise and sensitive. If relatively simple fluorescence intensity measurements of a solution needs to be performed, we will refer to these methods as simple fluorometric quantification, there are several technical possibilities, from easy to use single sample measurements using fluorometers like Qubit (Invitrogen) or Quantus (Promega), offering single- or dual- channel measurements, respectively, up to higher throughput microplate readers, like the Safire series (Tecan) which allows measurements in a continuous spectrum. When compared to spectrophotometry, fluorometry is not so convenient in the estimation of the presence of different contaminants, however, it is also less susceptible to concentration overestimations occurring due to the presence of such contaminations. In addition, it has a sensitivity to accurately measure even lower concentrations of cfNAs [271], while there are several assays for different concentration distributions (from tens of picograms up to hundreds of nanograms per microliter), as well as for different types of NAs (dsDNA, ssDNA, RNA, miRNA) [274]. Fluorometry based methods provide accurate quantification, but do not allow quantification of selective size range [275].

### 6.2. Sequence Nonspecific Methods Allowing Quantification and Sizing of cfNAs Fragments

Partly based on the same physical principles as simple fluorometry, i.e., staining of NAs through sequence nonspecific DNA binding fluorescent dyes, *capillary electrophoresis* is performed in a time-resolved manner during an electrophoretic separation of NAs fragments. Accurate quantification of cfNAs on automated capillary electrophoresis provides, therefore, highly precise analytical evaluation of various NAs, even with regard to the size of the fragments. Although capillary electrophoresis of sequence specific PCR amplicons, labeled with covalently bound fluorophores, on genetic analyzers may also be used as a main analytical method [267], sequence nonspecific quantification and sizing of cfNAs performed by capillary electrophoresis was shown to be sensitive and demonstrated good quantitation accuracy and reproducibility [270] moreover it is able to detect post-sampling cellular DNA contamination that may often be observed after cfNAs isolation [270]. Instruments like the Bioanalyzer [276] and TapeStation (Agilent, Santa Clara, CA, USA) [270,276], or Experion (Bio-Rad, Hercules, CA, USA) [277,278], for example, enable highly precise characterization of NAs concentrations and distribution of fragment sizes, however, only in certain detection ranges, which depends on the used consumables and kits. Since for DNA it is around from few picograms up to a thousand of picograms per microliter, while for RNA from tens of picograms up to thousands of picograms per microliter [279], it may happen that the concentration of cfDNA/cfRNA will not fall within this range, whereupon additional quantification by another method may be necessary [270]. Despite certain limitations, the spread of MPS put electrophoresis, together with simple fluorimetry-based quantification, to the center of sequencing library preparation procedures, and are therefore considered prerequisites for more complex analyses of cfNAs applications.

### 6.3. Sequence Specific Methods Allowing Low-Throughput Characterization of Single or Only a Few Genomic Loci per Assay

Conventional *polymerase chain reaction* (PCR), performed using fluorescently labelled oligonucleotide primers, coupled with an end-point detection of amplified products using capillary electrophoresis on genetic analyzers may be used for analyzes of cfNAs. It was reported to allow, for example, detection and genotyping of paternally inherited Y chromosome specific microsatellite alleles in cffDNA fragments of male fetuses present in maternal plasma [267]. Different modifications of PCR are, however, widely used among cfNAs analytical methods.

*Real-time quantitative PCR* (qPCR) is a widely used detection and quantification method, sensitive enough to work also with low amounts of nucleic acids, including cfDNA [238]. Employing one of the several detection techniques, from DNA binding fluorescent dyes up to sequence-specific fluorescently labelled probes, it allows continuous monitoring of the amplification progression [280]. Since the increase of the amplification product depends on the initial template copy number and on the amplification efficiency, qPCR is generally used for the detection of cfNAs fragments of interest, estimation of PCR inhibition [281], as well as for absolute or relative quantification using carefully selected reference genes [282,283]. With regard to quantification, however, it was reported that cfDNA quantities based on measurements of some target genes were, on average, more than twofold higher than those of other assays, suggesting that analysis and averaging of multiple reference may lead to more reliable estimates of total cfDNA quantity [238]. Since the used oligonucleotides are sequence specific, qPCR allows also to determine the origin of cfNAs, discriminating, for example, between fetal and maternal fragments [284], or between genomic and mitochondrial cfNAs [283].

*In droplet digital PCR* (ddPCR), sometimes regarded as a third generation of PCR technologies, target samples are partitioned, for example using oil droplets, allowing highly parallel but isolated amplification of single molecules of the same target DNA sequence, while the amplification itself, i.e., the presence or absence of the target molecule in the droplet is detected through fluorescent probes, but in and end-point manner [7]. Since it allows absolute quantification achieving orders of magnitude more precision and sensitivity than qPCR, ddPCR may be considered if quantification, but not complex size distribution estimation of NAs is required. In addition to quantification, for example of circulating fetal and maternal DNA from cell-free plasma [7], however, ddPCR was also used for determination of sample quality and the degree of cellular DNA contamination using differently sized amplicons of a single target region [285], or even of multiple target regions to ensure higher precision from limited quantities of cfDNA [286]. It should be mentioned here that significant differences were detected between archived and immediately processed samples, in the term of quality, when using ddPCR and comparing various extraction kits. On the other hand, comparing different collection protocols near no differences were detected in the term of quality and quantity [286].

Although conventional dideoxynucleotide-termination-based Sanger sequencing, i.e., one of the most commonly used first generation sequencing methods, may have a relatively common role in everyday clinical practice utilizing cfNAs, this method is not conventionally listed among the main cfNAs analytical technologies. It is used most commonly if the detection of certain sequences, or specific mutations, is of interest in cfNAs derived from specific body fluids, most typically in the field of oncogenetics [287].

### 6.4. Sequence Specific Methods Allowing Highly Parallel Genome-Wide Characterization of NAs

Even in the field of cfNAs applications, MPS based second and third generation sequencing methods are gaining more and more attention [288], especially since the housing of low-coverage whole genome sequencing with the recognition of the presence of cffDNA in maternal plasma. Following this they quickly became a popular and globally used testing strategy with rapidly increasing numbers of analyzes carried out annually [289,290,291]. Moreover, beyond NIPT, the use of this technique started to show not only secondary possibilities of NIPT data re-use for large-scale but low-cost population studies [5,6], but also an outbreak into unrelated clinical fields, such as oncology in a form of liquid biopsy-based non-invasive cancer diagnostics (NICD) [292].

Although both second and the third-generation sequencing is represented by many methods using different technological procedures, they generally work on the principles of miniaturized parallel sequencing of millions of DNA fragments. In second generation methods, fragments undergo clonal amplification during sequencing library preparation, while the sequencing length typically ranges from tens to hundreds of bases [293]. In contrast, third generation methods rely on single-molecule sequencing, i.e., they do not need clonal amplification, while they tend to have read lengths in several kilobases [294]. With regard to cfNAs applications, even in cases of fragments having different origins, such as fetal NAs in maternal plasma or cancer genomes in patient plasma, entire genomes can be characterized using MPS. Although using highly different read depths, genome-scale high-throughput screening and genotyping of the majority of DNA variant types was reported, from characterization of epigenetic markers of NAs [295,296], through SNVs and a wide range of CNVs [297,298,299], including even microsatellites [300] and chromosomal aneuploidies [301]. Moreover, highly sensitive MPS approaches may be further modified to achieve even higher sensitivity of detection and quantification of low frequency mutations in plasma, for example using modifications such as the Safe-Sequencing System (Safe-SeqS). This utilizes assignment of unique molecular identifiers to each template molecule, clonal amplification of each uniquely tagged template molecule, followed by redundant sequencing of the amplification products [13,302].

In addition, apart from the ability to detect a wide range of variations and modifications, the impact of MPS on cfNAs applications is further enhanced by its high sensitivity, but also by its ability to characterize both the ratios of cfNAs of different origins and the size distributions of the fragments. Determination of the ratios is based on the sequence composition itself and on the epigenetic landscape of the fragments, while their size distribution characterization may be based on insert sizes when using paired-end sequencing [297]. When estimating size ranges, however, specific steps of library preparation should be considered and possibly modified to prevent the loss of certain fragments during this highly selective process.

### 6.5. Less Commonly Used Methods for cfNAs Analyzes

Alongside with the above mentioned methods, although less commonly reported, there are also other analytical techniques for characterization of different aspects of cfNAs. For example for detection of low frequency mutations in tumor associated cfDNAs, high specificity and sensitivity was reported using the combination of common methods like high-resolution melting (HRM) analysis of PCR amplicons, restriction fragment length polymorphism (RFLP) assays and Sanger sequencing [287]. BEAMing, called on the basis of four principal components of the method (i.e., beads, emulsion, amplification, and magnetics) combines oligonucleotide-coupled-bead-based emulsion PCR amplification, single base extension and counting of fluorescently labeled particles via flow cytometry [303], allowing for highly sensitive detection and quantification of somatic mutations in the plasma of patients, for example suffering from colorectal tumors [304]. From the highly parallel genome-wide techniques, although high-throughput microarray-based assays (like array-based comparative genomic hybridization) are not so frequently used in cfNAs applications, examples of their use do also exist [129,305].

### 6.6. Concluding Remarks for cfNAs Analytical Methods

Fluorometric quantification using fluorescent dyes is an approach that compensates for some disadvantages of spectrophotometry-based quantification. Nevertheless, these two quantification methods allow the most effective, fast, very reliable and not much expensive cfNAs quantification that can be repeated many times from low amounts of input material. It is highly recommended to use the electrophoresis based method in parallel with fluorometric methods when quantification and sizing is required for the final evaluation of cfNAs profiles, or for downstream analyzes [177,275,306]. However, the detection range of some electrophoresis-based methods is very narrow and it is unlikely that the majority of samples will have similar concentration which will fall within its upper limit of quantification. The overestimated concentration may lead to chip overloading, which in turn provides inaccurate results, so this quantification is generally not suitable for first-line quantification of cfDNA [270]. Majority of laboratories prefer simple fluorometric quantification, which provides an easy workflow and saves time as compared to qPCR. qPCR is a sensitive, accurate and reproducible quantification method, however, it requires broad optimization of the PCR reaction conditions [177] or to obtain pre-optimized commercial qPCR assays which are generally much more expensive. ddPCR quantification is highly sensitive, reproducible and modern technology, but comparing the cost and time required for measurement is very convenient [307]. Despite several advantages, neither qPCR nor ddPCR is able to determine the size range of fragments, however, with certain modifications they may be used to estimate degradation status of the sample [285,286]. On the other hand, it is absolutely important to realize when designing and performing cfNAs analyzes that concentration estimates generally differ when using different methods, even for the same sample, mainly because of different principles and limitations of the methods, as well as because several factors may deteriorate results [268]. When comparing spectrophotometric, simple fluorometric (using PicoGreen) and qPCR-based (using SYBR Green I assay) quantification methods in the light of DNA fragmentation, although the overall sensitivity of the spectrophotometric measurement was found to be the lowest, this was the only method remaining unaffected by fragmentation of the measured fragments. Both the fluorometric and the qPCR-based assays were significantly influenced, with a decrease in measured concentration connected to more intensive DNA fragmentation, suggesting that DNA quantification should be performed using equally fragmented control DNAs [308]. When considering the sequencing methods, it is important to keep in mind that different sequencing platforms suffer from contrasting limitations and laboratory biases, making them suitable for different classes of problems. As we mentioned above, second generation sequencers generate high yields of relatively high-quality sequenced fragments that have established their dominant position in the direct detection of small genomic variations in human genomes, especially SNVs and small CNVs, or methylation patterns [309]. However, due to their short size, they are impractical for direct detection of large genomic variations, such as larger CNVs or other structural rearrangements [310]. When these types of variants need to be assessed from short read sequencing data, indirect statistical evaluation of target region coverage is required (for details see the next chapter and citations there), while this is impractical for the detection of balanced rearrangements. Third generation sequencers promise to alleviate this problem by sequencing of much longer fragments, albeit yet at the cost of much lower base quality. The appropriate platform should be, therefore, carefully selected according to the goals of the testing, while combination of several platforms may also offer the potential to employ their complementary advantages to improve overall results [311].

## 7. Data Analysis

Many of the abovementioned analytical methods rely on computational data collection, processing and evaluation, while in the majority of cases the relevant instruments are accompanied with dedicated commercial software. This is yet highly different in case of MPS. Although there are dedicated platform-associated bioinformatic tools too, extensive genome-scale data sets, having wide potential for further reanalysis and reuse also for other purposes than for which they were originally generated, promote the development of newer and newer bioinformatic tools and pipelines. As a result, bioinformatic tools for processing MPS data are still under intensive development and diversification [312]. On the other hand, an unprecedented amount of genomic data is generated every day, even for cfNAs, which poses a major challenge for computational and statistical methods to reveal usable patterns and answers to the diagnostic question, or to the experimental hypotheses. However, virtually all applications are based on sampling, laboratory processing, and measurements, which inevitably bias the generated data from the actual genomic content of the sample. Improperly prepared cfNAs may lead to unsuccessful method performance and consequently to the results misinterpretation. Therefore, the generated data should always be checked for known forms of distortions [313], and state-of-the-art methods should be applied to mitigate their effects and remove sequencing artifacts. Although some bias can be reduced without losing sequenced data [314,315], it is more common to remove sequencing artifacts such as low-quality base calls [316], remnants of sequencing adapters [316,317], contamination induced by laboratory processing [318], or artificial duplications induced by PCR amplification [319]. Since each such preprocessing step lengthens the computational analysis, makes it more complex, and therefore more prone to failure, and most importantly reduces the data set, refinements in the laboratory methods should be preferred to increase the yield of high-quality genomic data for subsequent analyzes.

In general, if germline variants of the analyzed individual are to be detected from his own cfNAs, they are present in a large portion of the sequenced fragments making them easier to detect. Conventional tools may be utilized for data processing, such as Bowtie 2 for alignment with subsequent specific normalization steps and, for example CNV calling on normalized bin counts using circular binary segmentation algorithm provided by the R package DNAcopy [6,299].

On the other hand, the great challenge is to uncover the indicative patterns in fragments that are present only in the small fraction of the DNA mixture. Since oncological diseases are typical with frequent somatic variations due to defects in genome maintenance [320], ranging from small variations [321] to even large structural variations [322] and methylation patterns [323], detection of unusually high, but low frequency variation among the sequenced fragments may be indicative [324,325]. Increasing sequencing depth is a widely used approach to improve mutation calling performance, which can be complemented by fine tuning of pipelines to ensure high specificity in identifying somatic variants supported by only a few reads, since variant caller tools such as Mutect [326], Varscan [327], Vardict [328], Strelka2 [329], may perform differently in different sequencing depth and variant frequency [330]. The problem of low frequency variants is similar also in non-invasive prenatal care, specifically in cases when germline sequence variants of a fetus are to be detected among the cfDNA fragments of the mother.

When identifying larger genomic variants, such as CNVs, derived for example from the fetal genome but present in the maternal plasma, detection methods generally does not rely on the direct identification of the variant in the reads, due to length limitations, rather on indirect assessment through the relative number of mapped reads to different genomic regions. These measurements are quantitative, while the aberrant number of DNA material from the genomic region indicates the duplication or deletion of the affected chromosomal segment. Similarly to maternal CNVs described above, sample normalization, read counting for individual bins and CNV calling was described to be efficient for the identification of fetal CNVs in cfDNA isolated from maternal plasma [299,331]. The main problem of this field is, however, still the relatively small proportion of fetal DNA present in the maternal blood, typically from five to 15% of the cfDNA mixture, that is most commonly alleviated by large amounts of sequenced DNA fragments that allow to more precisely distinguish the aberrant number of fragments from the affected fetal chromosome from the inaccuracies of measurements. A lot of effort has been put into developing more robust biostatistical methods to reduce the number of required sequenced fragments, and thus increase the viability of tests by statistical highlighting of fragments that are more likely to originate from the tissue of interest. So far, several characteristics with different separation capabilities have been described. The combination of these complementing characteristics into a single meta-predictor has great promise for improving overall accuracy and detecting new useful patterns in the sequenced fragments [332,333].

Fragment lengths are a well-studied aspect of the sequenced cfNA fragments in several clinical domains [334], since they tend to differ in different body fluids (Appendix A), but also with regard to the cells of origin and mechanisms of release [335]. Although length alone cannot be unambiguously used to distinguish fragments, it can be used to estimate the proportions of a DNA mixture [333] or to focus analysis on fragments belonging to selected size ranges using robust statistical modeling [331,332]. Length of cfDNA fragments may be determined as the difference of the leftmost and the rightmost mapped base of the corresponding read pair, following, for example, mapping using the Bowtie 2 algorithm [333,336].

Mapping location is also a powerful metric for finding relevant patterns. Although analysis focused only on a specific region is too unreliable, the combination of a signal along the entire genome can reveal useful patterns [337,338]. In addition, profiling end motifs and centering around nucleosomes may also indicate the tissue of origin, albeit typically with less accuracy [339].

Moreover, in addition to the high importance of germline and somatic genomic variation of the analyzed individual, a significant amount of genomic material sequenced from the sampled material cannot be directly attributed to the known human genome, i.e., remains unmapped following alignment to the human reference genome. This genomic “dark matter” typically represents the organisms present in the host, and so their sequencing represents a great opportunity to monitor the balance of the microbiome or to detect potentially harmful pathogens [340]. With the improvements of the sequencing technologies, algorithms, and statistical methods, along with ongoing efforts to capture genomic diversity of all living organisms, we can expect massive improvements in the detection of pathogens or harmful disbalances of the microbiome.

## 8. Conclusions

cfNAs are present in various body fluids, including those commonly assessed, such as blood plasma, saliva, or urine, but also in less commonly considered body fluids, which may be, on the other hand, in direct contact with hardly accessible tissues or organs. They may reflect genomic alterations, ongoing physiological processes as well as changes in the organism in response to various stimuli. Therefore, main biomedical applications consider and use them as biomarkers of several pathological conditions, but their role as potential therapeutic targets is also emerging. cfNAs content of many body fluids in certain diseases is still under intensive research, so they had limited or no use in clinical practice, but certainly not to the same extent as blood plasma. Although cfNAs represent promising material, their analyzes is not so easy with many different approaches existing, since standardized protocols for the processing of cfNAs are still missing. Moreover, currently available methods for analyzes require cfNAs input of sufficient quality and quantity to obtain reliable results, therefore individual steps of the whole workflow should be carefully considered and designed. As we described above, this workflow consists of several steps, including (i) sample collection, storage, and transport, (ii) processing of biological material, (iii) extraction of cfNAs, (iv) analytical process, and (v) data analysis and interpretation of the results. Each level of this workflow may be affected by environmental and technical factors, which should be considered, especially in the case of those applications where the collection site of the biological material is usually elsewhere than the processing laboratory. Since cfNAs present in plasma are the most studied, the majority of sampling devices, buffers, protocols for processing, or extraction kits are designed for the analysis of cfNAs originated from this source. For those less commonly analyzed body fluids, specialized collection devices and/or preservation solutions have not yet been developed. Because of these limitations, commercial assays developed for total DNA or RNA extraction from the matching body fluid, or assays dedicated to extraction of cfNAs but from other body fluids, are commonly reported in the literature.

Taken together, since each of the delineated steps and procedures may contain several more or less appropriate modules, the best choice to design the most appropriate combination will always depend on several factors, including the aims of the analyzes, application requirements, available budget, but also the technical and personal infrastructure of the workplaces involved in the analyzes. And last but not least, thorough validation of the whole complex process should always be performed before its implementation takes place.

## Figures and Tables

**Figure 1 ijms-21-08634-f001:**
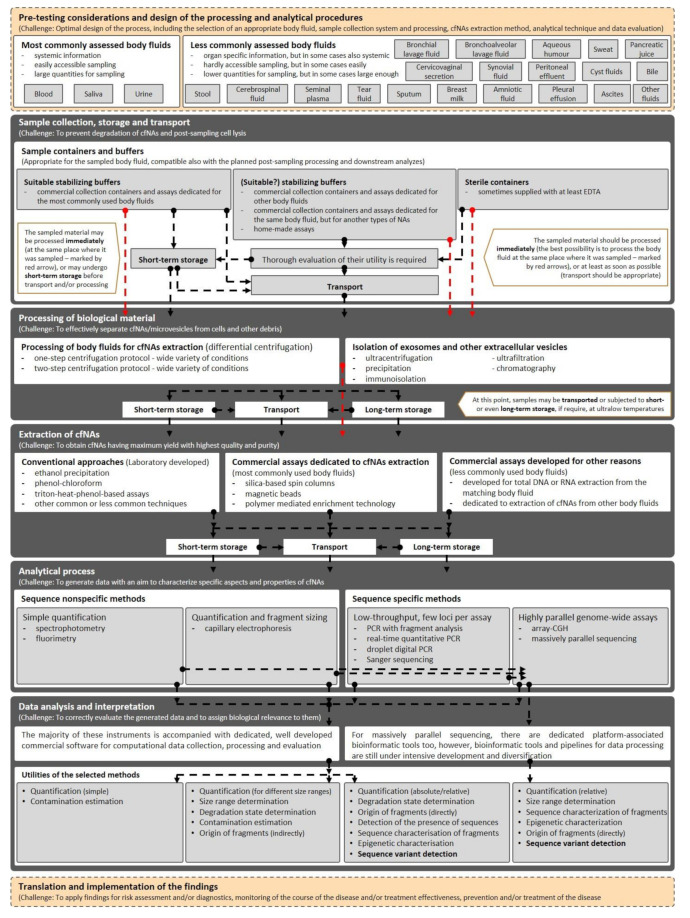
Schematic representation of the complex process of cfNAs analyzes divided according to the main steps and their most relevant individual parts and supplemented with specific notes and main points of consideration. Processing and extraction of cfNAs should be performed as soon as possible after the sampling, therefore, the best practice may be to perform the whole processing at the same place at which the body fluids were sampled. These inter-step connections are highlighted in the scheme by red arrows and dashed lines.

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
