# Peer review of "Technical and Methodological Aspects of Cell-Free Nucleic Acids Analyzes"

_ijms, 2020, doi:10.3390/ijms21228634_

Round 1

Reviewer 1 Report

The manuscript by Poes et al. is a very detailed and technical summary of methods used to analyze extracellular nucleic acids. The review begins with an introduction in to the topic of extracellular DNAs and RNAs and then follows by systematically reviewing different aspects of extracellular nucleic acid analysis, including different types of body fluids, sample preparation techniques, sample processing and storage, analytical methods, and data analysis methods. I must admit that the review is very detail-oriented and it was sometimes diffucult to "pounder" through the text, but I think it is a necessity in the case of such a technical review. I think the review will be a very useful compendium for people working or aiming to start working in the field of extracellular nucleic analysis. Therefore I recommend publication in the current form.

Author Response

Dear reviewer, we appreciate your time to spend reviewing our manuscript.Thank you for your positive feedback and for the recommendation to publish our manuscript in the current form.

Kind regards! 

Zuzana Pös

Reviewer 2 Report

Analyses of DNA, RNAs and subtypes of DNA, mtDNA, bacterial DNA, miRNAs, etc have shown promising potential in biomedical applications. Their analyzes comprise many steps, from sample collection, storage and transportation, and bioinformatic analyses and statistical evaluations.  Each of these steps has a potential to affect the outcome and informational value of the performed. Presently there are no standard protocols analyzing such samples.  Authors therefore prepared an overview by analyzing literature in an attempt to make uniform workflow, processes, and outcomes.

Critique

Manuscript is well-written and contain key information for those colleagues affiliated with steps from sample collection/preparation to analysis and data interpretations.  Sections are exhaustively discuss all aspects methodologies, potential outcomes and each of them has concluding remarks.  Therefore, an additional conclusion (section 8) seems to be repetition what already been said (it is two pages long).  Please make a much shorter conclusion, not exceeding or about half a page.

Author Response

Critique from the reviewer 2:

Manuscript is well-written and contain key information for those colleagues affiliated with steps from sample collection/preparation to analysis and data interpretations.  Sections are exhaustively discuss all aspects methodologies, potential outcomes and each of them has concluding remarks.  Therefore, an additional conclusion (section 8) seems to be repetition what already been said (it is two pages long).  Please make a much shorter conclusion, not exceeding or about half a page.

Response:

Dear reviewer, we appreciate your time to spend with the revision of our manuscript.Thank you for your positive feedback and little critique to shorten our conclusion. Based on your recommendation we edited section 8, from which we deleted information repeating from the previous sections which already included concluding remarks.

Kind regards!

Zuzana Pös